# Influence of Isocyanate Structure on Recyclable Shape Memory Poly(thiourethane)

**DOI:** 10.3390/ma16114040

**Published:** 2023-05-29

**Authors:** Yu Zeng, Jiale Song, Jinfu Li, Chi Yuan

**Affiliations:** Department of Polymer Materials and Engineering, School of Materials Science and Engineering, Chang’an University, Xi’an 710018, China; 2020131030@chd.edu.cn (Y.Z.); 20202310095@chd.edu.cn (C.Y.)

**Keywords:** poly(thiourethane), thermosets, shape memory, reprocess

## Abstract

In this study, poly(thiourethane) (PTU) with different structures is synthesized by click chemistry from trimethylolpropane tris(3-mercaptopropionate) (S3) and different diisocyanates (hexamethylene diisocyanate, HDI, isophorone diisocyanate, IPDI and toluene diisocyanate, TDI). Quantitative analysis of the FTIR spectra reveals that the reaction rates between TDI and S3 are the most rapid, resulting from the combined influence of conjugation and spatial site hindrance. Moreover, the homogeneous cross-linked network of the synthesized PTUs facilitates better manageability of the shape memory effect. All three PTUs exhibit excellent shape memory properties (R_r_ and R_f_ are over 90%), and an increase in chain rigidity is observed to negatively impact the shape recovery rate and fix rate. Moreover, all three PTUs exhibit satisfactory reprocessability performance, and an increase in chain rigidity is accompanied by a greater decrease in shape memory and a smaller decrease in mechanical performance for recycled PTUs. Contact angle (<90°) and in vitro degradation results (13%/month for HDI-based PTU, 7.5%/month for IPDI-based PTU, and 8.5%/month for TDI-based PTU) indicate that PTUs can be used as long-term or medium-term biodegradable materials. The synthesized PTUs have a high potential for applications in smart response scenarios requiring specific glass transition temperatures, such as artificial muscles, soft robots, and sensors.

## 1. Introduction

Shape memory polymers (SMPs) are intelligent materials that possess a unique ability to revert from a temporary shape to their original and permanent shape under some specific conditions, such as temperature variations [1], water [2], chemicals [3], and light [4], among other factors. They can be widely used in the fields of aerospace, biomedicine, 4D printing, and flexible robots. Based on their chemical composition, SMPs can be categorized into physically cross-linked thermoplastic SMPs and covalent cross-linked thermosetting SMPs. Thermoplastic SMPs exhibit impressive recoverable strains (>700%) [5]; however, their physical networks are prone to creep and irreversible plastic deformation, which can decrease shape stability and recovery rate [6]. In comparison, the thermoset SMPs demonstrate a superior shape fixation rate and recovery rates, as well as excellent thermal and chemical stability, owing to stable covalent cross-linked networks [7,8]. Consequently, the body of research pertaining to them is significantly more extensive. The thermosetting SMPs can be considered as an ideal alternative for the next generation of shape memory materials.

Recently, polyurethane (PU) has become one of the most extensively reported thermoset SMPs owing to low cost, wide range of monomer options, and diverse chain segment structure [9]. Shape memory PU (SMPU) can be utilized in a myriad of applications, spanning from compression bandages [10,11] to medical implants [12,13]. Ahmad et al. synthesized a class of SMPU to apply in compression bandages, which can be activated by heat to tailor pressure [10]. Narayana et al. synergistically incorporated SMPU filament with nylon to fabricate intelligent medical stockings, capable of offering static pressure, therapeutic massage effects, and convenient size adjustment, thereby enabling smart compression therapy [11]. The SMPU/hydroxyapatite/reduced graphene oxide/arginyl-glycyl-aspartic acid composites with remarkable cell adhesion were fabricated by Zhang and colleagues, which can be implemented in invasive bone repair [12]. Wang et al. synthesized water-based PU as the main component of a 3D printing ink for fabricating bone scaffolds. Superparamagnetic iron oxide nanoparticles (SPIO NPs) were incorporated into the ink to promote osteogenic induction and shape fixity [13]. However, the broad applications and commercial success have brought about the generation and steady increase in PU waste, including end-of-life and post-consumer products [14]. Because of the presence of contaminants and deformities, PU waste, especially from thermosetting PU, is less likely to be reused. Therefore, it is necessary to develop recyclable PU-based SMPs to modify structures.

Poly(thiourethane)s (PTUs) are a class of polymers with similar structure and properties to PU. In PTU, the dynamic covalent bonds within the cross-linking framework can bring about its topological alterations without impacting the degree of cross-linking, allowing PTU to undergo remolding, self-welding, and reprocessing [15]. Huang et al. reported that PTU could be depolymerized into oligomers after introducing thiourethane (TU) bonds into the cross-linked network. The depolymerized oligomers could cross-link into PTU again without loss of mechanical properties [16]. Li et al. demonstrated that the exchange reactions occur between the TU group and the free thiol group at high temperature to thermally convert into thiols and isocyanates. The optimized cross-linked PTU elastomers synthesized via the chemical method can ensure complete recovery of their cross-link density and tensile properties, even after undergoing multiple rapid remodeling cycles [17]. Fan et al. successfully synthesized a PTU–PU network exhibiting self-healing properties at a moderate temperature and reprocessing properties at a higher temperature [18]. Moreover, the synthesis of PTU is less complex owing to click chemistry reactions and does not involve too many by-reactions [19,20]. Even so, the research on the shape memory properties of PTU remains relatively limited. Some researchers have prepared a mixed polymer network by introducing TU bonds into other shape memory polymer networks. Yue et al. developed a thermoset shape memory PTU by using isocyanate-terminated polyurethane to react with pentaerythritol tetrakis (3-mercaptopropionate), introducing multiple dynamic covalent bonds to enhance its recyclability [21]. Zhou et al. fabricated a PU–PTU network with an excellent thermoadaptive reversible two-way shape memory effect [22]. Nguyen et al. synthesized a PU–PTU network with shape memory properties and self-healing ability via Diels–Alder chemistry [23]. Some researchers have used isocyanate to polymerize with thiol directly to form shape memory PTU. Wang et al. fabricated a smart and shape-reconfigurable wood by incorporating PTU into delignified wood scaffolds. The resulting transparent wood exhibited remarkable reconfigurable shape memory behavior, a superior optical property, and low thermal conductivity [24]. Dailing et al. prepared a PTU network through UV initiation and then used a small stoichiometric excess of thiol groups on the surface to fabricate dexamethasone-loaded nanogel through a UV-initiated thiol–ene reaction [25]. Cui et al. employed a reactive diluent methyl methacrylate and dynamic cross--linker prepared from 2,2′-(ethylenedioxy)-diethanethioland and 2-isocyanatoethyl methacrylate as printing ink to fabricate 4D printing PTU [26]. Although different kinds of PTUs have been developed, the influence of different isocyanate structures on the properties of PTU is still a blank.

In this work, various PTUs were synthesized using trimethylolpropane tris(3-thiolpropionate) in combination with different diisocyanates, including hexamethylene diisocyanate, isophorone diisocyanate, and toluene diisocyanate. The chemical structure and thermal stability of PTUs were measured by Fourier transform infrared (FTIR) and thermogravimetric analysis (TGA). The shape memory property of PTUs was evaluated by dynamic mechanical analysis (DMA). Additionally, the shape memory performance of reformed PTU samples by milled and hot-pressed process was also tested. Finally, the water contact angle and in vitro degradation of PTUs were measured.

## 2. Materials and Methods

### 2.1. Materials

Trimethylolpropane tris(3-thiolpropionate) (S3), hexamethylene diisocyanate (HDI), isophorone diisocyanate (IPDI), toluene diisocyanate (TDI), dibutyltin dilaurate (DBTDL), and phosphate buffered saline (PBS) powder were purchased from Aladdin Reagent Co., Ltd., Shanghai, China.

### 2.2. Sample Preparation

The diisocyanate and S3 were added in a 3:2 molar ratio and the specific usage of each component is listed in Table 1. Additionally, a 1% amount of DBTDL was pre-dissolved in S3 to ensure uniform distribution. The mixture was then poured into a PTFE mold with dimensions of 20 × 5 × 1 mm^3^ and molded at a temperature of 120 °C.

### 2.3. Characterization

FTIR spectra were carried out by using a Vertex 70 Fourier Transform Infrared (Bruker, Billerica, MA, USA) spectrometer equipped with an attenuated total reflection (ATR) accessory. The molecular structures of HDI, IPDI, and TDI were molded in Chem3D software, subsequently subjecting these structures to geometric optimization via the MOPAC/AM1 technique to acquire stable conformations. Gaussian09W computational software was used to determine the Mulliken charge distributions for the diisocyanate compounds, utilizing the B3LYP calculate method to achieve comprehensive optimization and establish the basis group as 6-31G. The thermal stability of the samples was assessed via TGA using a TGA2 (Mettler Toledo, Zurich, Sweden) at a rate of 10 ℃/min under nitrogen atmosphere. The thermomechanical properties of the materials were analyzed using a DMA Q800 (TA Instruments, New Castle, DE, USA) equipped with a 3-point bending clamp. The experiments were carried out at a heating rate of 3 ℃/min, from 30 to 160 ℃, at 1 Hz and 0.1% of strain. The shape memory performance of the samples was analyzed through DMA Q800 (TA Instruments, New Castle, DE, USA) in controlled force mode. The recycled samples were produced by grinding the cross-linked polymers and hot-pressing them into an aluminum mold at a temperature of 170 °C for 3 h under a pressure of 8 MPa. Then, the shape memory characteristic curve of the recycled sample was tested by DMA Q800 (TA Instruments, New Castle, DE, USA). Water contact angles of PTUs were determined using the sessile drop method, and surfaces were cleaned with anhydrous ethanol. A micro-injector was used to inject 1–5 μL water droplets from a position at 2–3 mm above the test surface, forming a comprehensive three-phase contact line between solid and liquid surfaces after 5–10 s. The conical cone method was employed to calculate water contact angles. A standard phosphate-buffered saline solution, simulating body fluids, was prepared and utilized for observing degradation behavior at room temperature.

## 3. Results

The possible reaction route of fabricating PTU is illustrated in Figure 1. In the reaction, the TU group was formed by the addition reaction between the thiol group and the isocyanate group, where the hydrogen atom in the thiol group was incorporated into the nitrogen atom of the isocyanate group. The TU group bore a resemblance to the urethane group; in contrast, the oxygen atom of the urethane group was replaced by a sulfur atom. To characterize the as-prepared PTUs, the FTIR spectra of the three isocyanate-thiol reaction systems were measured before and after the reaction, as shown in Figure 2. The absorption peak observed at 2270 cm^−1^ belonged to the stretching vibration of the isocyanate group (-NCO). The absorption peak at 3350 cm^−1^ demonstrated the presence of -N-H, and the absorption peak at 1670 cm^−1^ was assigned to the carbonyl bond [27,28]. After the additional reaction, the three-reaction system revealed a reduction in the isocyanate group signal and the emergence of the N-H bond and carbonyl bond new peaks. The above result indicated that the PTU was successfully synthesized. Moreover, there were the residual isocyanate groups after the reaction, suggesting the non-completed conversion from isocyanate group and thiol group to TU bond in thermal curing. The conversion rate of the isocyanate groups was measured to evaluate the extent of thiol–isocyanate reaction. To obtain the conversion of isocyanate groups, a quantitative analysis was performed on the peaks of isocyanate groups in the FTIR spectra at different moments throughout the entire reaction process. Figure 3a–c shows the trend of the isocyanate group during the reaction. Through the quantification of isocyanate peak intensities and the following percentage reduction of the peak area, the conversion rate of the isocyanate group was determined by Equation (1).
(1)$$x_{NCO}=\frac{A_{2270}}{A_{2270,0}}$$
where *A*_2270_ and *A*_2270,0_ represent the absorbances of the isocyanate peak at 2270 cm^−1^ at a given reaction time and at the beginning of the curing process, respectively, after being normalized. The corresponding conversion rate profiles of various isocyanates can be observed in Figure 3d. It is shown that the order of reaction rate with S3 was TDI > HDI > IPDI.

Mulliken charge distribution calculations serve as a valuable tool for elucidating the underlying mechanisms of chemical reactions and the intricate interplay between atoms. By providing insights into the distribution of electron density within a molecular system, these calculations contribute to a deeper understanding of the fundamental processes governing chemical transformations and the intricate nature of atomic interactions. Quantum chemical calculations yielded Mulliken charge distributions for various structural diisocyanates, as shown in Figure 4. Table 2 lists charge values associated with nitrogen, carbon, and oxygen atoms present in the distinct isocyanate groups.

The molecular structure of TDI included a benzene ring to which two isocyanate groups were directly connected. The π-conjugation of the benzene ring established an extensive conjugation system with the isocyanate groups, consequently diminishing the electron cloud density within TDI’s isocyanate groups and intensifying the positive charge of carbon atoms in isocyanate groups, particularly in 2,6-TDI. The thiol–isocyanate addition reaction, being a nucleophilic addition, benefited from the reduction in the electron cloud density of the isocyanate groups, promoting the reaction rate between the thiol and isocyanate groups. While the benzene ring exhibited considerable spatial hindrance, the electron conjugation effect surpassed the ring’s spatial hindrance, rendering TDI the most reactive among the three examined diisocyanates.

The IPDI molecule featured a cyclohexyl group, an isocyanate group directly linked to the cyclohexyl group, and another isocyanate group connected to the same carbon atom of the cyclohexyl group with a methylene group. As the cyclohexyl group functioned as an electron-donating group, it elevated the electron cloud density of the carbon atoms in the isocyanate group, particularly the isocyanate group directly connected to the cyclohexyl group. This enhancement increased the electronegativity, which was unfavorable for the thiol–isocyanate nucleophilic addition, exhibiting reduced reactivity. Simultaneously, the cyclohexyl group’s spatial hindrance was unfavorable for the isocyanate group’s mobility, leading to a decreased likelihood of effective collisions with the thiol group, ultimately rendering the thiol’s nucleophilic anionic attack the slowest.

In the HDI molecule, the isocyanate groups were linked to a weakly electron-donating aliphatic chain, resulting in the electropositivity of carbon atoms in the isocyanate group exhibiting a medium order between TDI and IPDI. The aliphatic methylene carbon chain possessed high flexibility and minimal spatial resistance, thus rendering the reactivity of this system in a medium position.

DMA afforded an indirect insight into the uniformity of the cross-linked network and enable the determination of the glass transition temperature (*T_g_*) of the samples. Figure 5a,b shows the energy storage modulus and loss factor of the three samples. It was shown that the *T_g_* of HDI-based PTU was approximately 60 °C, whereas the *T_g_* of IPDI-based PTU and TDI-based PTU were approximately 130 °C. The enhanced *T_g_* was attributed to the elevation of chain rigidity. Additionally, the tan δ peaks of all PTUs were quite narrow, and the half-peak widths did not exceed 15 °C. This phenomenon was attributed to the homogeneous cross-linked network formed through the click chemistry reaction. Furthermore, the transition from the glassy to the rubbery state was notably sharp and concentrated for all three samples. These characteristics endowed the PTUs with an exceedingly narrow range of trigger temperatures for the shape memory effect, leading to the easier control of shape memory effect. Furthermore, to maintain the proper working of the shape memory material, the thermal stability of different PTUs was also measured to analyze the thermal degradation characteristics, as shown in Figure 5c. The thermal decomposition processes of the three PTUs were consistent. From the characteristic degradation temperature (*T_5%_*), the thermal degradation temperature of the three PTUs exceeded the respective *T_g_* (shape memory triggering temperature), which is the shape memory triggering temperature, implying that shape memory effects can be initiated within the service temperature range.

The shape memory performances of three PTUs were assessed via DMA. When the temperature was elevated to above *T_g_* (the strain, 𝜀_𝑋_ = 0), a consistent load was imposed to induce strain. The load was maintained to cool down below *T_g_*, and the strain at this moment was recorded as 𝜀_(*𝑌*,*load*)_. After this, the load was released to record the strain at this moment as 𝜀_𝑌_. Then the temperature was elevated above *T_g_* to recover the strain, and the strain at this moment was recorded as 𝜀_(𝑋,*rec*)_. Equations (2) and (3) were used to compute the shape fix rate (*R_f_*) and the shape recovery rate (*R_r_*).
(2)$$R_{f}=\frac{\varepsilon_{Y}-\varepsilon_{X}}{\varepsilon_{\left(Y,load\right)}-\varepsilon_{X}}$$
(3)$$R_{r}=\frac{\varepsilon_{Y}-\varepsilon_{\left(X,rec\right)}}{\varepsilon_{Y}-\varepsilon_{X}}$$

The obtained stress-strain-temperature diagrams are shown in Figure 6a–c, and the corresponding shape fix rate *R_f_* and *R_r_* of the samples are summarized in Table 3. The applied load of the HDI-based PTU was lower than that of the IPDI-based PTU and TDI-based PTU because of its lower strength. It was noteworthy that all samples exhibited a high shape recovery rate. In detail, the shape memory performance (such as *R_r_* and *R_f_*) was HDI > IPDI > TDI. The difference may be attributed to the chain rigidity. In the reduced flexible chain, the constrained mobility within the cross-linked network restricted shape recovery, leading to the inferior shape memory performance of the SMP. The shape memory trigger temperature of all three samples was correlated with *T_g_* from the test data of the thermomechanical properties, implying that the elastic modulus of chain recovery from the glass to the rubber state is the driving force of the shape memory effect of PTU.

More visually, taking HDI-based PTU as a representative, Figure 6d–f presents the photographic representation for the SME test of the PTU sample. The temporary shape of the sample was achieved by folding at 90 °C and rapidly cooling to room temperature. Then, the fixed sample reverted to the permanent shape when the temperature rose again to 90 °C. The visual demonstration of the SME test highlighted the remarkable shape memory performances of HDI-based PTUs, which exhibited high shape recovery ratios and excellent mechanical properties.

In addition to its shape memory properties, the PTU can be reprocessed because of its dynamic covalent bonding, even cross-linked thermosetting PTUs. The recycling performance of the PTU is exhibited in Figure 7a–c. The sample was ground into powder, added into a hot press mold, and then pressed at 7 MPa and 160 °C for 3 h to the recycled sample. The transparency of the original sample was acceptable, while the transparency of the recovered sample was slightly reduced and exhibited a slight yellowing effect. Furthermore, the shape memory properties of the recycled PTUs were subsequently evaluated via DMA and are presented in Figure 7d–f. The corresponding *R_f_* and *R_r_* of the samples are also summarized in Table 3. Compared with the original PTUs, the recycled PTUs exhibited more significant strain under the same stress, reducing tensile modulus. The HDI-based PTU with the aliphatic group displayed a more substantial decrease, nearly 40%, while the TDI-based PTU with the aromatic group showed a relatively lesser decline, at nearly 30%. The variation in this behavior was attributed to the chain rigidity, where higher rigidity confers better resistance to the damage on the bond. Furthermore, apart from the dynamical broken-linked TU bonds, the grinding process can also affect other bonds; thus, the decline in shape memory performance was acceptable. A minor reduction in the trigger temperature of the shape memory effect was also observed, resulting from the cross--linked network structure alterations caused by the grinding-hot-pressing process.

The prospective biocompatibility and biodegradability of PTU have been reported [29,30,31]; water contact angles and in vitro degradation tests with three synthesized PTUs were evaluated to ascertain their potential in biomaterial applications. Similar to PUs, PTUs can form hydrogen bonds with water, rendering them hydrophilic. Water contact angles for the three PTUs, as shown in Figure 8, followed the order IPDI-based PTU > TDI-based PTU > HDI-based PTU. This was attributed to the IPDI-based PTU having a higher conversion rate, resulting in a larger molecular weight, and the presence of a non-polar alicyclic ring, which increases the contact angle despite the addition of TU bonds. The non-polar benzene ring in TDI elevated the contact angle of the TDI-based PTU greater than the HDI-based PTU, while the HDI-based PTU exhibited the smallest contact angle because of its short aliphatic chain. Nevertheless, all three exhibited water contact angles of less 90°, showing hydrophilicity; thus, it can be tentatively concluded that all three PTUs are biocompatible.

In vitro degradation tests were conducted to further explore the potential of PTUs in biological applications. PTUs were sliced into 5 mm × 5 mm × 1 mm pieces and individually weighed. Samples were taken from PBS solution at regular intervals every day. After vacuum drying for 2 h, they were weighed to determine the degradation rate of the PTUs. Degradation test results are presented in Figure 9. Biodegradation rates for the three curing systems were 13%/month for HDI-based PTU, 7.5%/month for IPDI-based PTU, and 8.5%/month for TDI-based PTU. PTUs have the potential to be used as medium-term or long-term biodegradable materials. Degradation rates followed the order HDI-based PTU > TDI-based PTU > IPDI-based PTU, consistent with the hydrophilicity trends observed in the contact angle test. The results showed that the higher the hydrophilicity of the PTU, the easier it was to be infiltrated by PBS solution, and thus the higher the degradation rate.

## 4. Conclusions

In this study, we successfully synthesized reprocessable PTUs with the shape memory effect through commercially available aliphatic and aromatic isocyanates, ternary thiols, and organotin catalysts. Quantitative analysis of the FTIR spectra at different moments revealed that the reaction rates between diisocyanates and S3 followed the order TDI > HDI > IPDI. The reaction rate order were explained through quantum chemical calculations (Mulliken charge distributions). Moreover, the homogeneous cross-linked network of the synthesized PTUs facilitated better manageability of the shape memory effect. All three PTUs exhibited excellent shape memory properties (*R_r_* and *R_f_* were over 90%), and the triggering temperature corresponded to the respective glass transition temperature. An increase in chain rigidity was observed to negatively impact the shape recovery rate and fix rate of all three materials. Moreover, all three materials exhibited satisfactory recycle performance, a decrease in tensile modulus of 30–40%, and shape memory properties of about 5% were observed. An increase in chain rigidity was accompanied by a greater decrease in shape memory and a smaller decrease in mechanical performance for the recycled PTUs. In addition, water contact angles and in vitro degradation tests were conducted to explore the potential of PTUs in biological applications. All three PTUs exhibited biocompatibility (water contact angle <90°) and biodegradability. Biodegradation rates for the three curing systems were 13%/month for HDI-based PTU, 7.5%/month for IPDI-based PTU, and 8.5%/month for TDI-based PTU. These PTUs can be used as medium-term or long-term biodegradable materials. Nonetheless, the excellent mechanical and thermal properties of the three PTUs, along with the impressive recycling performance even under challenging conditions, enable their utilization in smart response scenarios that necessitate specific glass transition temperatures, such as artificial muscles, soft robots, and sensors.

## Figures and Tables

**Figure 1 materials-16-04040-f001:**
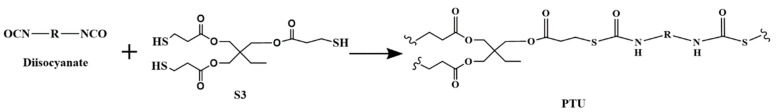
Schematic representation of the thiol–isocyanate possible reaction route between S3 and diisocyanate.

**Figure 2 materials-16-04040-f002:**
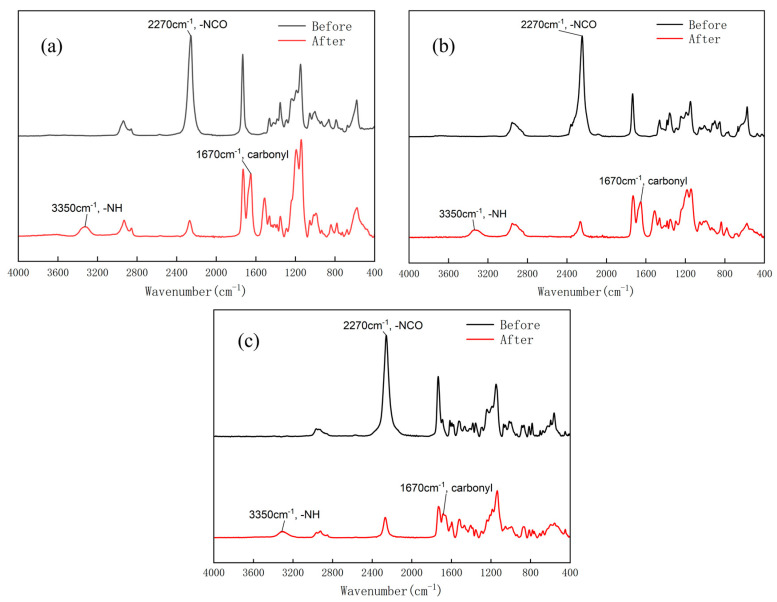
FTIR spectra of (**a**) HDI-based PTU, (**b**) IPDI-based PTU, and (**c**) TDI-based PTU before and after the reaction.

**Figure 3 materials-16-04040-f003:**
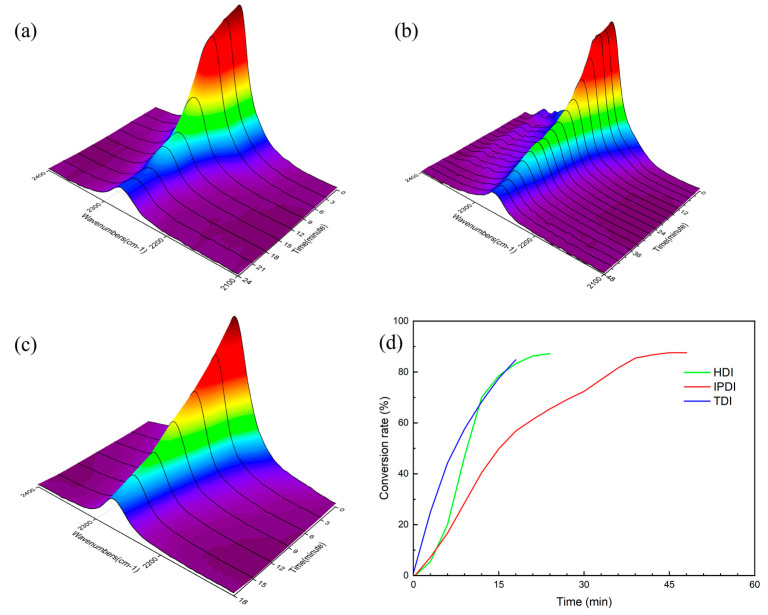
(**a**) HDI, (**b**) IPDI, and (**c**) TDI NCO group conversion trends with time and (**d**) the three diisocyanate NCO conversion rates with time.

**Figure 4 materials-16-04040-f004:**
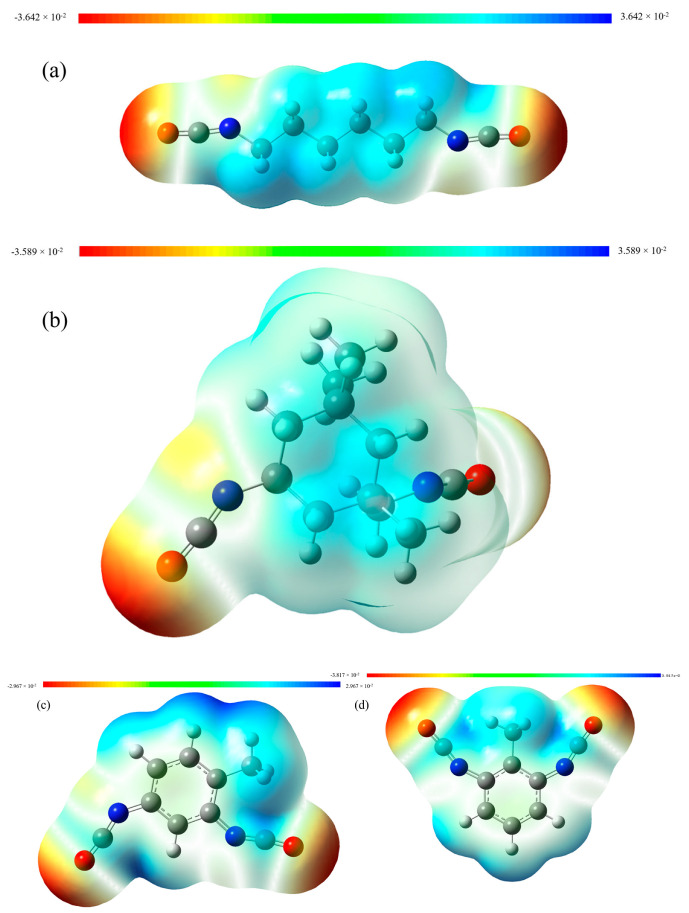
The charge distribution figure of (**a**) HDI (**b**) IPDI (**c**) 2,4-TDI (**d**) 2,6-TDI.

**Figure 5 materials-16-04040-f005:**
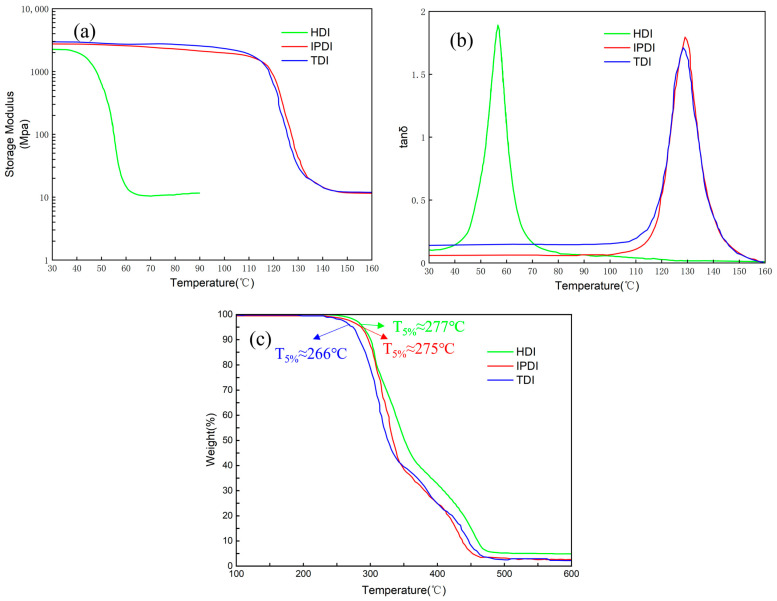
(**a**) Tan δ and (**b**) storage modulus evolution with temperature and (**c**) TGA curves for the different PTUs prepared.

**Figure 6 materials-16-04040-f006:**
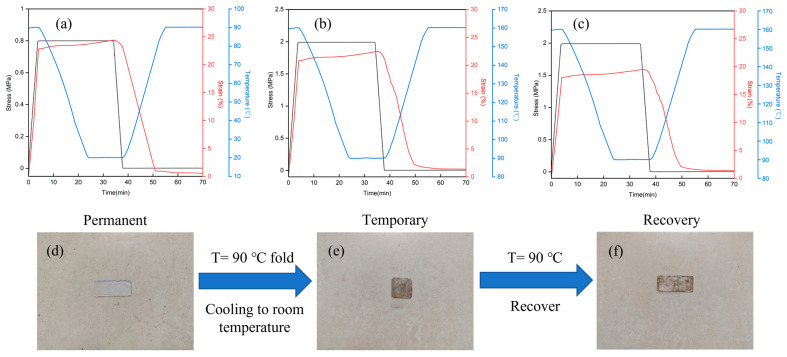
Shape memory characteristic curve of the (**a**) HDI-based PTU, (**b**) IPDI-based PTU, and (**c**) TDI-based PTU and photographs of (**d**) the permanent shape, (**e**) temporary shape, (**f**) recovery shape.

**Figure 7 materials-16-04040-f007:**
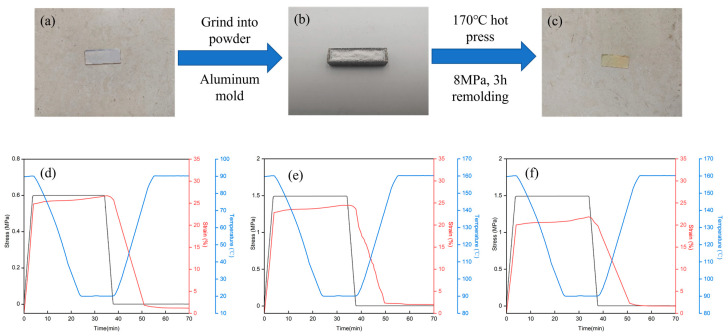
Photographs of the (**a**) original, (**b**) grinded, and (**c**) recycled HDI-based PTU and the shape memory characteristic curve of the (**d**) recycled HDI-based PTU, (**e**) recycled IPDI-based PTU, and (**f**) recycled TDI-based PTU.

**Figure 8 materials-16-04040-f008:**
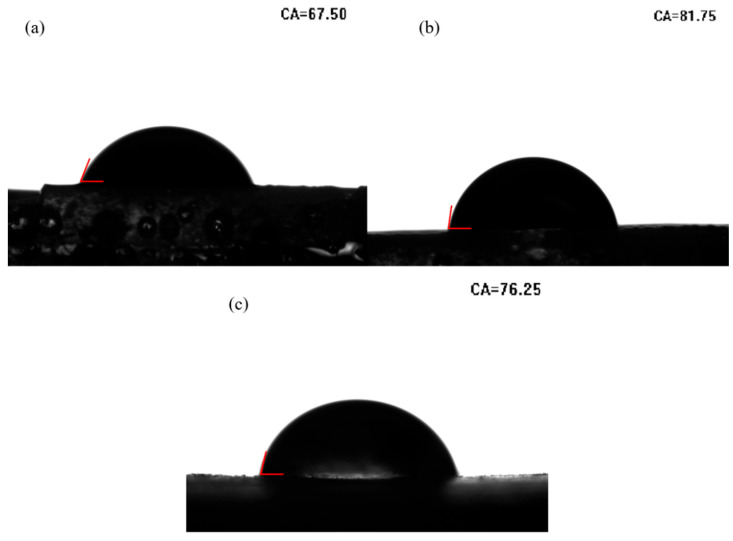
Water contact angles of (**a**) HDI-based PTU, (**b**) IPDI-based PTU, and (**c**) TDI-based PTU.

**Figure 9 materials-16-04040-f009:**
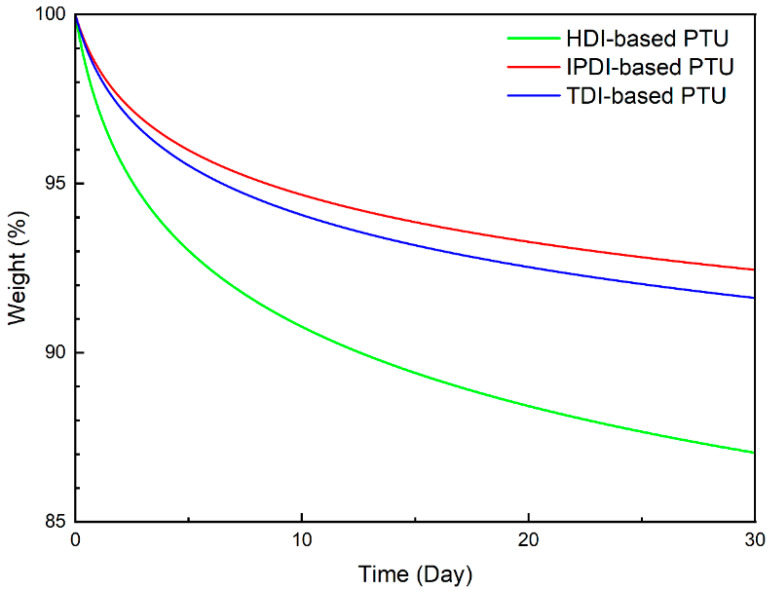
Fitted degradation curves of three PTUs in PBS solution, 30 days.

**Table 1 materials-16-04040-t001:** The usages of HDI, IPDI, TDI, and S3.

Sample	Weight	
	Diisocyanate	S3
HDI + S3	1.69 g	4.69 g
IPDI + S3	3.37 g	
TDI + S3	1.78 g	

**Table 2 materials-16-04040-t002:** Mulliken charge distribution of N, C, and O in isocyanate groups with different chain structures.

Diisocyanate	N1	C1	O1	N2	C2	O2
HDI	−0.430	0.603	−0.412	−0.430	0.603	−0.412
IPDI	−0.430	0.595	−0.415	−0.420	0.697	−0.410
2,4-TDI	−0.494	0.588	−0.400	−0.505	0.603	−0.392
2,6-TDI	−0.649	0.750	−0.495	−0.652	0.773	−0.486

**Table 3 materials-16-04040-t003:** Shape memory test data.

Sample	*R_f_* (%)	*R_r_* (%)
HDI-based PTU	96.9	97.7
IPDI-based PTU	93.4	95.1
TDI-based PTU	92.2	93.2
Recycled HDI-based PTU	95.5	95.3
Recycled IPDI-based PTU	92.0	91.8
Recycled TDI-based PTU	89.7	91.5

## Data Availability

The data presented in this study are available on request from the corresponding author.

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
