# Peer review of "Influence of Isocyanate Structure on Recyclable Shape Memory Poly(thiourethane)"

_materials, 2023, doi:10.3390/ma16114040_

Round 1

Reviewer 1 Report

This paper clearly articulates the shape memory effect of the TPUs and is suitable for publication in the materials journal. Figures 3, 4, 5, 6 and 7 have blurred lines, and some axis labels are unclear. I recommend redrawing the plots in those Figures to improve their quality.

Some clarity issues are highlighted below

Line 119- remove ‘a duration of’

Line 206 -  remove ‘basically’

Line 207 - ……can exceed respective Tg (shape memory triggering temperature), implying ….

Line 236 - rapidly cooling down room temperature  change to rapidly cooling to room temperature

Line 237 – remove ‘back’

Line 252 -253 : Change to the following

Compared with the original PTUs, recycled PTUs exhibit more significant strain under the same stress, reducing tensile modulus.

Line 269-260: Change to

A minor reduction in the trigger temperature of the shape memory effect is also observed, resulting from the cross-linked network structure alterations caused by the grinding-hot pressing process.

Line 312 – 313: Change to

All three PTUs exhibited biocompatibility and biodegradability and could be used as medium-term or long-term biodegradable materials.

Reviewer 2 Report

The manuscript by Zeng et al. investigates a series of self-synthesized poly(thiourethanes) having shape memory properties. This is an interesting work, which fits well with materials. However, the work has significant weaknesses and should therefore be improved. Here are my suggestions:

-         The abstract lacks specific results. Please revise accordingly. 

-         Literature on polyurethane-based SMPs is not balanced very well.

-         Pls introduce an abbreviation for PTU in the abstract and specify smart response scenarios.

-         The resolution of Figures 6 and 7 should be improved. Compared with other shape memory polymers the maximum strain applied is very low. Please describe material behavior when larger strains are applied.

-         With regard to folding experiments, please quantify the fixed angle / recovery angle and introduce ratios as done in equations 2 and 3. Pls add a discussion by comparing the results of folding with those of stretching.

-         Pls add information regarding FT-IR spectroscopy to gain a deeper understanding on what exactly happened during recycling. Pls add discussion.   

-         The designation of the samples, e.g. in Table 3 is misleading. HDI, for example, has no shape memory properties.

-         Line 239: Pls replace rates by ratios.

-         The conclusions are a summary of the findings presented and should therefore be rewritten. Where are current limitations?

-

Reviewer 3 Report

This manuscript presented a study about the influence of isocyanate structure on recyclable shape memory poly(thiourethane). The work has some potential. However, the authors must considered improve the discussions about the effect of isocyanate structure on the properties evaluated. Several comments and suggestions are described below.

Section 2.2: Was the authors used 1% of DBTDL in weight (1 wt%)?

Section 2.2: What was the pressure and time used to molded the  mixture?

Section 2.3: please better describe the parameters conditions (number of scans, scan range,….) used in FTIR analysis.

Section 3 and Figure 1: I suggest to the authors in figure captions of Figure 1 and the beginning of section 3 adopt “possible reaction route”.

Section 3: I suggest add references that can confirm that bands at 2270, 3350 and 1670 cm-1 are related to –NCO, -N-H and carbonyl, respectively.

Figure 2: improve figure quality and mark the main bands discussed in the main text on the figure. I suggest better discuss the results presented in Figure 2 a, b and c.

Section 3 lines 151-152: Why the order of reaction rate with S3 is TDI > HDI >IPDI? Please better explain this result.

Section 3 lines 196-197: Please better explain this affirmation: “The enhanced Tg is attributed to the elevation of chain rigidity”. Which may contribute to “elevation of chain rigidity? What is the effect of TDI, HDI, IPDI on this result?

Table 3: what is the standard deviation for all values presented in Table 3?

Figure 6: improve figure quality. It is very hard to see the details in Figure 6 d, e and f.

Figure 7: improve figure quality. It is very hard to see the details in Figure 7 a, b and c.

Lines 268-273: I suggest add the details used to determine the contact angle and section 2.

Figure 8: what is the standard deviation of the contact angles presented in Figure 8.

There are two Figure 8 in the manuscript. Please check.

Minor editing of English language required.

Round 2

Reviewer 2 Report

-

-

Reviewer 3 Report

After corrections the manuscript reads well. I suggest publication in its current form. 

The language requires minor editing.